# TRAINING DEEP NEURAL-NETWORKS USING A NOISE ADAPTATION LAYER

**Jacob Goldberger & Ehud Ben-Reuven**
Engineering Faculty, Bar-Ilan University,
Ramat-Gan 52900, Israel
`jacob.goldberger@biu.ac.il,udi.benreuven@gmail.com`

## ABSTRACT

The availability of large datsets has enabled neural networks to achieve impressive recognition results. However, the presence of inaccurate class labels is known to deteriorate the performance of even the best classifiers in a broad range of classification problems. Noisy labels also tend to be more harmful than noisy attributes. When the observed label is noisy, we can view the correct label as a latent random variable and model the noise processes by a communication channel with unknown parameters. Thus we can apply the EM algorithm to find the parameters of both the network and the noise and estimate the correct label. In this study we present a neural-network approach that optimizes the same likelihood function as optimized by the EM algorithm. The noise is explicitly modeled by an additional softmax layer that connects the correct labels to the noisy ones. This scheme is then extended to the case where the noisy labels are dependent on the features in addition to the correct labels. Experimental results demonstrate that this approach outperforms previous methods.

## 1 INTRODUCTION

The presence of class label noise inherent to training samples has been reported to deteriorate the performance of even the best classifiers in a broad range of classification problems (Nettleton et al. (2010), Pechenizkiy et al. (2006), Zhu & Wu (2004)). Noisy labels also tend to be more harmful than noisy attributes (Zhu & Wu (2004)). Noisy data are usually related to the data collection process. Typically, the labels used to train a classifier are assumed to be unambiguous and accurate. However, this assumption often does not hold since labels that are provided by human judgments are subjective. Many of the largest image datasets have been extracted from social networks. These images are labeled by non-expert users and building a consistent model based on a precisely labeled training set is very tedious. Mislabeling examples have been reported even in critical applications such as biomedical datasets where the available data are restricted (Alon et al. (1999)). A very common approach to noisy datasets is to remove the suspect samples in a preprocessing stage or have them relabeled by a data expert (Brodley & Friedl (1999)). However, these methods are not scalable and may run the risk of removing crucial examples that can impact small datasets considerably.

Variants that are noise robust have been proposed for the most common classifiers such as logistic-regression and SVM (Frénay & Verleysen (2014), Jakramate & Kabán (2012), Beigman & Klebanov (2009)). However, classifiers based on label-noise robust algorithms are still affected by label noise. From a theoretical point of view, Bartlett et al. (2006) showed that most loss functions are not completely robust to label noise. Natarajan et al. (2013) proposed a generic unbiased estimator for binary classification with noisy labels. They developed a surrogate cost function that can be expressed by a weighted sum of the original cost functions, and provided asymptotic bounds for performance. Grandvalet & Bengio (2005) addressed the problem of missing labels that can be viewed as an extreme case of noisy label data. They suggested a semi-supervised algorithm that encourages the classifier to predict the non-labeled data with high confidence by adding a regularization term to the cost function. The problem of classification with label noise is an active research area. Comprehensive up-to-date reviews of both the theoretical and applied aspects of classification with label noise can be found in Frénay & Kaban (2014) and Frénay & Verleysen (2014).

In spite of the huge success of deep learning there are not many studies that have explicitly attempted to address the problem of Neural Net (NN) training using data with unreliable labels. Larsen et al. (1998) introduced a single noise parameter that can be calculated by adding a new regularization term and cross validation. Minh & Hinton (2012) proposed a more realistic noise model that depends on the true label. However, they only considered the binary classification case. Sukhbaatar & Fergus (2014) recently proposed adding a constrained linear layer at the top of the softmax layer, and showed that only under some strong assumptions can the linear layer be interpreted as the transition matrix between the true and noisy (observed) labels and the softmax output layer as the true probabilities of the labels. Reed et al. (2014) suggested handling the unreliability of the training data labels by maximizing the likelihood function with an additional classification entropy regularization term.

The correct unknown label can be viewed as a hidden random variable. Hence, it is natural to apply the EM algorithm where in the E-step we estimate the true label and in the M-step we retrain the network. Several variations of this paradigm have been proposed (e.g. Minh & Hinton (2012), Bekker & Goldberger (2016)). However, iterating between EM-steps and neural network training does not scale well. In this study we use latent variable probabilistic modeling but we optimize the likelihood score function within the framework of neural networks. Current noisy label approaches assume either implicitly or explicitly that, given the correct label, the noisy label is independent of the feature vector. This assumption is probably needed to simplify the modeling and derive applicable learning algorithms. However, in many cases this assumption is not realistic since a wrong annotation is more likely to occur in cases where the features are misleading. By contrast, our framework makes it easy to extend the proposed learning algorithm to the case where the noise is dependent on both the correct label and the input features. In the next section we describe a model formulation and review the EM based approach. In Section 3 we described our method which is based on adding another softmax layer to the network and in Section 4 we present our results.

## 2 A PROBABILISTIC FRAMEWORK FOR NOISY LABELS

Assume we want to train a multi-class neural-network soft-classifier $p(y = i|x; w)$ where $x$ is the feature vector, $w$ is the network parameter-set and $i$ is a member of the class-set $\{1, ..., k\}$. We further assume that in the training process we cannot directly observe the correct label $y$. Instead, we only have access to a noisy version of it denoted by $z$. Here we follow the probabilistic modeling and the EM learning approach described in Bekker & Goldberger (2016). In this approach noise generation is assumed to be independent of the features and is modeled by a parameter $\theta(i, j) = p(z = j|y = i)$. The noise distribution is unknown and we want to learn it as part of the training phase. The probability of observing a noisy label $z$ given the feature vector $x$ is:

$$p(z = j|x; w, \theta) = \sum_{i=1}^{k} p(z = j|y = i; \theta)p(y = i|x; w) \tag{1}$$

where $k$ is the number of classes. The model is illustrated in the following diagram:

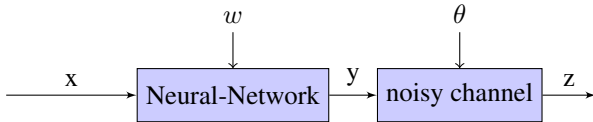

In the training phase we are given $n$ feature vectors $x_1, ..., x_n$ with the corresponding noisy labels $z_1, ..., z_n$ which are viewed as noisy versions of the correct hidden labels $y_1, ..., y_n$. The log-likelihood of the model parameters is:

$$L(w, \theta) = \sum_{t=1}^{n} \log(\sum_{i=1}^{k} p(z_t|y_t = i; \theta)p(y_t = i|x_t; w)) \tag{2}$$

Based on the training data, the goal is to find both the noise distribution $\theta$ and the Neural Network parameters $w$ that maximize the likelihood function. Since the random variables $y_1, ..., y_n$ are hidden, we can apply the EM algorithm to find the maximum-likelihood parameter set. In the E-step of

each EM iteration we estimate the hidden true data labels based on the noisy labels and the current parameters:

$$c_{ti} = p(y_t = i | x_t, z_t; w_0, \theta_0), \qquad i = 1, ..., k, \qquad t = 1, ..., n \qquad (3)$$

where $w_0$ and $\theta_0$ are the current parameter estimations. In the M-step we update both the NN and the noisy channel parameters. The updated noise distribution has a closed-form solution.

$$\theta(i,j) = \frac{\sum_t c_{ti} 1_{\{z_t=j\}}}{\sum_t c_{ti}}, \qquad i, j \in \{1, ..., k\} \qquad (4)$$

The $k \times k$ matrix $\theta$ can be viewed as a confusion matrix between the soft estimates of the true label $\{c_{ti} | i = 1, ..., k\}$ and the observed noisy labels $z_t$. As part of the EM M-step, to find the updated NN parameter $w$ we need to maximize the following function:

$$S(w) = \sum_{t=1}^{n} \sum_{i=1}^{k} c_{ti} \log p(y_t = i | x_t; w) \qquad (5)$$

which is a soft-version of the likelihood function of the fully observed case, based on the current estimate of the true labels. The back-propagation derivatives of the function (5) that we maximize in the M-step are:

$$\frac{\partial S}{\partial u_i} = \sum_{t=1}^{n} (p(y_t = i | x_t, z_t; w_0, \theta_0) - p(y_t = i | x_t; w)) h(x_t) \qquad (6)$$

such that $h$ is the final hidden layer and $u_1, ..., u_k$ are the parameters of the soft-max output layer.

The method reviewed here is closely related to the work of Minh & Hinton (2012). They addressed the problem of mislabeled data points in a particular type of dataset (aerial images). The main difference is that in their approach they assumed that they do not learn the noise parameter. Instead they assume that the noise model can be separately tuned using a validation set or set by hand. Note that even if the true noise parameters are given, we still need the apply the EM iterative procedure. However, this assumption makes the interaction between the E-step and the NN learning much easier since each time a data-point $x_t$ is visited we can compute the $p(y_t = i | x_t, z_t)$ based on the current network parameters and the pre-defined noise parameters. Motivated by the need for model compression, Hinton et al. (2014) introduced an approach to learn a "distilled" model by training a more compact neural network to reproduce the output of a larger network. Using the notation defined above, in the second training stage they actually optimized the cost function: $S(w) = \sum_{t=1}^{n} \sum_{i=1}^{k} p(y_t = i | x_t; w_0, \theta_0) \log p(y_t = i; x_t; w)$ such that $w_0$ is the parameter of the larger network that was trained using the labels $z_1, ..., z_n$, $w$ is the parameter of the smaller network and $\theta_0(i, j)$ in this case is a non-informative distribution (i.e. $\theta_0(i, j) = 1/k$).

There are several drawbacks to the EM-based approach described above. The EM algorithm is a greedy optimization procedure that is notoriously known to get stuck in local optima. Another potential issue with combining neural networks and EM direction is scalability. The framework requires training a neural network in each iteration of the EM algorithm. For real-world, large-scale networks, even a single training iteration is a non-trivial challenge. Moreover, in many domains (e.g. object recognition in images) the number of labels is very large, so many EM iterations are likely to be needed for convergence. Another drawback of the probabilistic models is that they are based on the simplistic assumption that the noise error is only based on the true labels but not on the input features. In this study we propose a method for training neural networks with noisy labels that successfully addresses all these problems.

## 3 TRAINING DEEP NEURAL NETWORKS USING A NOISE ADAPTATION LAYER

In the previous section we utilized the EM algorithm to optimize the noisy-label likelihood function (2). In this section we describe an algorithm that optimizes the same function within the framework of neural networks. Assume the neural network classifier we are using is based on non-linear intermediate layers followed by a soft-max output layer used for soft classification. Denote the non-linear

function applied on an input $x$ by $h = h(x)$ and denote the soft-max layer that predicts the true $y$ label by:

$$p(y = i|x; w) = \frac{\exp(u_i^\top h + b_i)}{\sum_{l=1}^{k} \exp(u_l^\top h + b_l)}, \qquad i = 1, ..., k \tag{7}$$

where $w$ is the network parameter-set (including the softmax layer). We next add another softmax output layer to predict the noisy label $z$ based on both the true label and the input features:

$$p(z = j|y = i, x) = \frac{\exp(u_{ij}^\top h + b_{ij})}{\sum_l \exp(u_{il}^\top h + b_{il})} \tag{8}$$

$$p(z = j|x) = \sum_i p(z = j|y = i, x)p(y = i|x) \tag{9}$$

We can also define a simplified version where the noisy label only depends on the true label; i.e. we assume that labels flips are independent of $x$:

$$p(z = j|y = i) = \frac{\exp(b_{ij})}{\sum_l \exp(b_{il})} \tag{10}$$

$$p(z = j|x) = \sum_i p(z = j|y = i)p(y = i|x) \tag{11}$$

We denote the two noise modeling variants as the complex model (c-model) (8) and the simple model (s-model) (10). Hereafter we use the notation $w_{\text{noise}}$ for all the parameters of the second softmax layer which can be viewed as a noise adaptation layer.

In the training phase we are given $n$ feature vectors $x_1, ..., x_n$ with corresponding noisy labels $z_1, ..., z_n$ which are viewed as noisy versions of the correct hidden labels $y_1, ..., y_n$. The log-likelihood of the model parameters is:

$$S(w, w_{\text{noise}}) = \sum_t \log p(z_t|x_t) = \sum_t \log(\sum_i p(z_t|y_t = i, x_t; w_{\text{noise}})p(y_t = i|x_t; w)) \tag{12}$$

Since the noise is modeled by adding another layer to the network, the score $S(w, w_{\text{noise}})$ can be optimized using standard techniques for neural network training. By setting

$$p(z = j|y = i) = \theta(i, j) = \frac{\exp(b_{ij})}{\sum_l \exp(b_{il})}, \tag{13}$$

it can easily verified that, by using either the EM algorithm (2) or the s-model neural network scheme (12), we are actually optimizing exactly the same function. Thus the neural network with the s-model noise adaptation layer provides an alternative optimization strategy to the EM algorithm. Instead of alternating between optimizing the noisy model and the network classifier, we consider them as components of the same network and optimize them simultaneously.

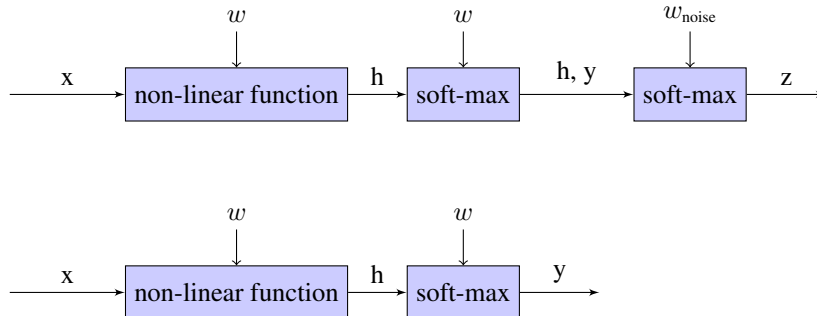

Figure 1: An illustration of the noisy-label neural network architecture for the training phase (above) and test phase (below).

Note that in the c-model, where the noise is also dependent on the input features, we can still apply the EM algorithm to learn the parameters of the additional noise layer. However, there is no closed-form solution in the M-step for the optimal parameters and we need to apply neural-network training in the M-step to find the noise-layer parameters.

At test time we want to predict the true labels. Hence, we remove the last softmax layer that aims to get rid of the noise in the training set. We compute the true-label softmax estimation $p(y = i|x; w)$ (7). The proposed architecture for training the neural network based on training data with noisy labels is illustrated in Figure 1.

There are degrees of freedom in the two softmax layer model. Hence, a careful initialization of the parameters of the noise adaptation layer is crucial for successful convergence of the network into a good classifier of the correct labels at test time. We used the parameters of the original network to initialize the parameters of the s-model network that contains the noise adaptation level. We can initialize the softmax parameters of the s-model by assuming a small uniform noise:

$$b_{ij} = \log((1 - \epsilon)1_{\{i=j\}} + \frac{\epsilon}{k-1}1_{\{i \neq j\}})$$

such that $k$ is the number of different classes. A better procedure is to first train the original NN without the noise-adaptation layer, ignoring the fact that the labels are noisy. We can then treat the labels produced by the NN as the true labels and compute the confusion matrix on the train set and used it as an initial value for the bias parameters:

$$b_{ij} = \log(\frac{\sum_t 1_{\{z_t=j\}}p(y_t = i|x_t)}{\sum_t p(y_t = i|x_t)})$$

such that $x_1, ..., x_n$ are the feature vectors of the training dataset and $z_1, ..., z_n$ are the corresponding noisy labels. So far we have concentrated on parameter initialization for the s-model. The strategy that works best to initialize the c-model parameters is to use the parameters that were optimized for the s-model. In other words we set linear terms $u_{ij}$ to zero and initialize the bias terms $b_{ij}$ with the values that were optimized by the s-model.

The computational complexity of the proposed method is quadratic in the size of the class-set. Suppose there are $k$ classes to predict, in this case the proposed methods require $k+1$ sets of softmax operations with a size of $k$ each. Hence there are scalability problems when the class set is large. As we explained in the previous paragraph, we initialized the second soft-max layer using the confusion matrix of the baseline system. The confusion matrix is a good estimation of the label noise. Assume the rows of the matrix correspond to the true labels and the matrix columns correspond to the noisy labels. The $l$ largest elements in the $i$-th row are the most frequent noisy class values when the true class value is $i$. We can thus connect the $i$-th element in the first softmax layer only to its $l$ most probable noisy class candidates. Note that if we connect the $i$-th label in the first softmax only to the $i$-th label in the second softmax layer, the second softmax layer collapses to identity and we obtain the standard baseline model. Taking the $l$ most likely connections to the second softmax layer, we allow an additional $l-1$ possible noisy labels for each correct label. We thus obtain a data driven sparsifying of the second softmax layer which solves the scalability problem since the complexity becomes linear in the number of classes instead of quadratic. In the experiment section we show that by using this approach there is not much deference in performance.

Our architecture, which is based on a concatenation of softmax layers, resembles the hierarchical softmax approach Morin & Bengio (2005) that replaces the flat softmax layer with a hierarchical layer that has the classes as leaves. This allowed them to decompose calculating the probability of the class into a sequence of probability calculations, which saves us from having to calculate the expensive normalization over all classes. The main difference between our approach and theirs (apart from the motivation) is that in our approach the true-label softmax layer is fully connected to the noisy-label layer. Sukhbaatar & Fergus (2014) suggested adding a linear layer to handle noisy labels. Their approach is similar to our s-model. In their approach, however, they proposed a different learning procedure.

## 4 EXPERIMENTS

In this section, we evaluate the robustness of deep learning to training data with noisy labels with and without explicit noise modeling. We first show results on the MNIST data-set with injected label

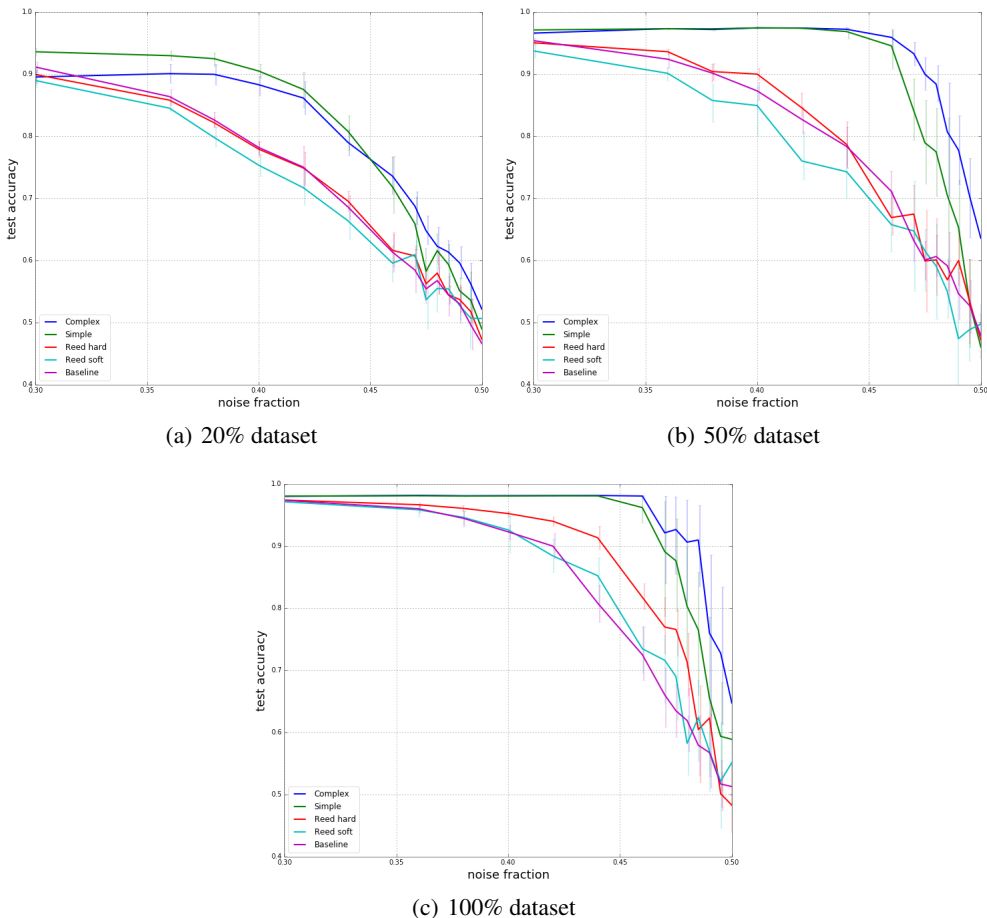

(a) 20% dataset

(b) 50% dataset

(c) 100% dataset

Figure 2: Test classification accuracy results on the MNIST dataset as a function of the noise level. The results are shown for several training data sizes (20%,50%,100%) of the training subset.

noise in our experiments. The MNIST is a database of handwritten digits, which consists of $28 \times 28$ images. The dataset has 60k images for training and 10k images for testing. We used a two hidden layer NN comprised of 500 and 300 neurons. The non-linear activation we used was ReLU and we used dropout with parameter 0.5. We trained the network using the Adam optimizer (Kingma & Ba (2014)) with default parameters, which we found to converge more quickly and effectively than SGD. We used a mini-batch size of 256. These settings were kept fixed for all the experiments described below. In addition to a network that is based on fully connected layers, we also applied a network based on a CNN architecture. The results we obtained in the two architectures were similar. The network we implemented is publicly available [1].

We generated noisy data from clean data by stochastically changing some of the labels. We converted each label with probability $p$ to a different label according to a predefined permutation. We used the same permutation as in Reed et al. (2014). The labels of the test data remained, of course, unperturbed to validate and compare our method to the regular approach.

We compared the proposed noise robust models to other model training strategies. The first network was the baseline approach that ignores the fact that the labels of the training data are unreliable. Denote the observed noisy label by $z$ and the softmax decision by $q_1, ..., q_k$. The baseline log-likelihood score (for a single input) is:

$$S = \sum_i 1_{\{z=i\}} \log(q_i)$$

---

[1]code available at `https://github.com/udibr/noisy_labels`

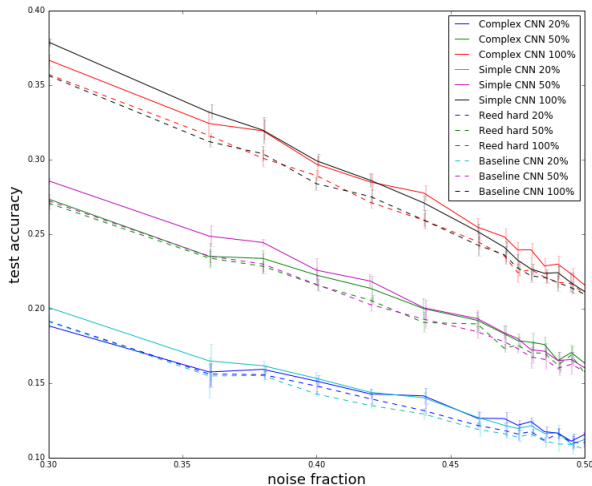

Figure 3: Test classification accuracy results on the CIFAR-100 dataset as a function of the noise level. The results are shown for several training data sizes (20%,50%,100%) of the training subset for a CNN network architecture).

We also implemented two variants of the noise robust approach proposed by Reed et al. (2014). They suggested a soft version

$$\beta S - (1-\beta)H(q) = \beta \sum_i 1_{\{z=i\}} \log(q_i) + (1-\beta) \sum_i q_i \log(q_i)$$

and a hard version:

$$\beta S + (1-\beta) \max_i \log(q_i)$$

In their experiments they took $\beta = 0.8$ for the hard version and $\beta = 0.95$ for the soft version, and observed that the hard version provided better results. Finally we implemented the two variants of our approach; namely, the noise modeling based only on the labels (s-model) and the noise modeling that was also based on the features (c-model).

Figure 2 depicts the comparative test errors results as a function of the fractions of noise. The results are shown for three different sizes of training data i.e. (20%,50%,100%) of the MNIST training subset. Bootstrapping was used to compute confidence intervals around the mean. For 1000 times, $N = 10$ samples were randomly drawn with repeats from the $N$ available samples and mean was computed. The confidence interval was taken to be the 2.5% and 97.5% percentiles of this process.

The results show that all the methods that are explicitly aware of the noise in the labels are better than the baseline which is the standard training approach. We revalidated the results reported in Reed et al. (2014) and showed that the hard version of their method performs better than the soft version. In all cases our models performed better than the alternatives. In most cases the c-model was better than the s-model. In the case where the entire dataset was used for training, we can see from the results that there was a phase transition phenomenon. We obtained almost perfect classification results until the noise level was high and there was a sudden strong performance drop. Analyzing why this effect occurred is left for future research.

We next show the results on the CIFAR-100 image dataset Krizhevsky & Hinton (2009) which consists of $32 \times 32$ color images arranged in 100 classes containing 600 images each. There are 500 training images and 100 testing images per class. We used raw images directly without any pre-processing or augmentation. We generated noisy data from clean data by stochastically changing some of the labels. We converted each one of the 100 labels with probability $p$ to a different label according to a predefined permutation. The labels of the test data remained, of course, unperturbed to validate and compare our method to the regular approach. We used a CNN network with two convolutional layers combined with ReLU activation and max-pooling, followed by two fully connected layers. Figure 3 depicts the comparative test errors results as a function of the fractions of noise for three different sizes of training data i.e. (20%,50%,100%) of the CIFAR-100 training

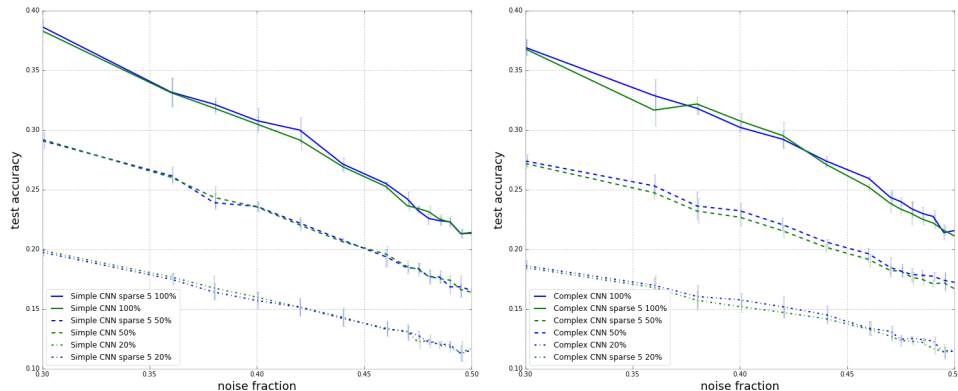

Figure 4: Test classification accuracy results on the CIFAR-100 dataset as a function of the noise level. The results of regular and sparse second softmax layers are shown for several training data sizes (20%,50%,100%) of the training subset .

subset. Bootstrapping was used to compute confidence intervals around the mean in the same way as for the MNIST experiment. The results showed that the proposed method works better than the alternatives. The simple model consistently provided the best results but when the noise level was very high the complex method tended to perform better.

We next report experimental results for the sparse variant of our method that remains efficient even when the class set is large. We demonstrate this on the case of the CIFAR-100 dataset which consists of 100 possible classes. For each class we only took the five most probable classes in the confusion matrix which is used to initialize the model parameter (see Section 3). As can be seen in Figure 4, sparsifying the second softmax layer did not not result in a drop in performance

## 5 CONCLUSION

In this paper we investigated the problem of training neural networks that are robust to label noise. We proposed an algorithm for training neural networks based solely on noisy data where the noise distribution is unknown. We showed that we can reliably learn the noise distribution from the noisy data without using any clean data which, in many cases, are not available. The algorithm can be easily combined with any existing deep learning implementation by simply adding another softmax output layer. Our results encourage collecting more data at a cheaper price, since mistaken data labels can be less harmful to performance. One possible future research direction would be to generalize our learning scheme to cases where both the features and the labels are noisy. We showed results on datasets with small and medium sized class-sets. Future research direction would be to evaluate the performance and efficiency of the proposed method on tasks with large class-sets.

ACKNOWLEDGMENTS

This work is supported by the Intel Collaborative Research Institute for Computational Intelligence (ICRI-CI).

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
