# Peer review of "Training deep neural-networks using a noise adaptation layer"

_ICLR 2017 — accepted_

[Official Review · AnonReviewer2 · rating 7 · confidence 5 · 16 Dec 2016]
**Interesting paper but lack of experiments**

The paper addressed the erroneous label problem for supervised training. The problem is well formulated and the presented solution is novel. 

The experimental justification is limited. The effectiveness of the proposed method is hard to gauge, especially how to scale the proposed method to large number of classification targets and whether it is still effective.

For example, it would be interesting to see whether the proposed method is better than training with only less but high quality data. 

From Figure 2, it seems with more data, the proposed method tends to behave very well when the noise fraction is below a threshold and dramatically degrades once passing that threshold. Analysis and justification of this behavior whether it is just by chance or an expected one of the method would be very useful.

[Official Review · AnonReviewer1 · rating 5 · confidence 5 · 16 Dec 2016]
**Training with Noisy Labels**

This work address the problem of supervised learning from strongly labeled data with label noise. This is a very practical and relevant problem in applied machine learning.  The authors note that using sampling approaches such as EM isn't effective, too slow and cannot be integrated into end-to-end training. Thus, they propose to simulate the effects of EM by a noisy adaptation layer, effectively a softmax, that is added to the architecture during training, and is omitted at inference time. The proposed algorithm is evaluated on MNIST and shows improvements over existing approaches that deal with noisy labeled data.

A few comments.
1. There is no discussion in the work about the increased complexity of training for the model with two softmaxes. 

2. What is the rationale for having consecutive (serialized) softmaxes, instead of having a compound objective with two losses, or a network with parallel losses and two sets of gradients?

3. The proposed architecture with only two hidden layers isn't not representative of larger and deeper models that are practically used, and it is not clear that shown results will scale to bigger networks. 

4. Why is the approach only evaluated on MNIST, a dataset that is unrealistically simple.

[Official Review · AnonReviewer3 · rating 5 · confidence 4 · 19 Dec 2016]
**This paper investigates how to make neural nets be more robust to noise in the labels**

This paper looks at how to train if there are significant label noise present.
This is a good paper where two main methods are proposed, the first one is a latent variable model and training would require the EM algorithm, alternating between estimating the true label and maximizing the parameters given a true label.

The second directly integrates out the true label and simply optimizes the p(z|x).

Pros: the paper examines a training scenario which is a real concern for big dataset which are not carefully annotated.
Cons: the results on mnist is all synthetic and it's hard to tell if this would translate to a win on real datasets.

- comments:
Equation 11 should be expensive, what happens if you are training on imagenet with 1000 classes?
It would be nice to see how well you can recover the corrupting distribution parameter using either the EM or the integration method. 

Overall, this is an OK paper. However, the ideas are not novel as previous cited papers have tried to handle noise in the labels. I think the authors can make the paper better by either demonstrating state-of-the-art results on a dataset known to have label noise, or demonstrate that a method can reliably estimate the true label corrupting probabilities.

[Author Response · Jacob Goldberger · 27 Dec 2016 (modified: 01 Jan 2017)]
**Reply to reviews**

We thank the reviewers for the comments and suggestions

1) We uploaded a revised version that contains experiments on CIFAR-100 dataset. We results on the CIFAR-100 are consistent with results on other data set and show that the performance of our method is better than previous methods.   

2) In the uploaded revised version we also addressed the scalability issue of the method in case of many classes. In our approach we initialized the second soft-max layer using the confusion matrix of the baseline system. The confusion matrix is a good estimation of the label noise.  Assume the rows of the matrix correspond to the true labels and the matrix columns correspond to the noisy labels. The k largest elements in the i-th row are the most occurring noisy class values when the true class value is i.    We can thus connect  i-th elements in the first softmax layer only to its k most probable noisy class  candidates.  (Note that if we connect the i-label in the first softmax only to the i-label in the second softmax layer,  we obtain the standard baseline model.) Taking only k connections to the second softamx layer solves the scalability problem. In the revised version we show experiments on CIFAR-100  that show that by using this scalable approach there is no performance degradation.

[Final Decision · Program Chairs · 06 Feb 2017]
**ICLR committee final decision**

Reviewers agreed that the problem was important and the method was interesting and novel. The main (shared) concerns were preliminary nature of the experiments and questions around scalability to more classes. 
 
 During the discussion phase, the authors provided additional CIFAR-100 results and introduced a new approximate but scalable method for performing inference. I engaged the reviewers in discussion, who were originally borderline, to see what they thought about the changes. R2 championed the paper, stating that the additional experiments and response re: scalability were an improvement. On the balance, I think the paper is a poster accept.